# “Looking beyond Mental Health Stigma”: An Online Focus Group Study among Senior Undergraduate Nursing Students

**DOI:** 10.3390/ijerph20054601

**Published:** 2023-03-05

**Authors:** Olga Valentim, Lídia Moutinho, Carlos Laranjeira, Ana Querido, Catarina Tomás, João Longo, Daniel Carvalho, João Gomes, Tânia Morgado, Tânia Correia

**Affiliations:** 1CINTESIS@RISE, Nursing School of Porto (ESEP), 4200-450 Porto, Portugal; 2Higher School of Nursing of Lisbon (ESEL), Av. Prof. Egas Moniz, 1600-096 Lisboa, Portugal; 3Nursing Research, Innovation and Development Centre of Lisbon (CIDNUR), Av. Prof. Egas Moniz, 1600-096 Lisboa, Portugal; 4Centre for Innovative Care and Health Technology (ciTechCare), Polytechnic of Leiria, Rua de Santo André—66–68, Campus 5, 2410-541 Leiria, Portugal; 5School of Health Sciences, Polytechnic of Leiria, Campus 2, Morro do Lena, Alto do Vieiro, Apartado 4137, 2411-901 Leiria, Portugal; 6Comprehensive Health Research Centre (CHRC), University of Évora, 7000-801 Évora, Portugal; 7Ribeiro Sanches Higher School of Health (ERISA)–IPLUSO, 1950-396 Lisboa, Portugal; 8Unidade de Investigação e Desenvolvimento em Educação e Formação (UIDEF), Instituto de Educação da Universidade de Lisboa, 1649-013 Lisboa, Portugal; 9Núcleo de Investigação em Ciências e Tecnologias da Saúde (NICITeS), Instituto Politécnico da Lusofonia, 1950-369 Lisboa, Portugal; 10Hospital Center of Leiria–Hospital de Santo André, R. de Santo André, 2410-197 Leiria, Portugal; 11Pediatric Hospital, Coimbra Hospital and University Centre, R. Dr. Afonso Romão, 3000-602 Coimbra, Portugal; 12The Health Sciences Research Unit: Nursing (UICISA: E), Nursing School of Coimbra (ESEnfC), 3004-011 Coimbra, Portugal

**Keywords:** mental health, social discrimination, focus groups, nursing students, stigma

## Abstract

Evidence highlights the need for professionals to be aware of their stigmatizing attitudes and discriminatory practices in order to minimize the negative impact on the people they take care of. However, nursing students’ perceptions of these issues have been poorly studied. This study explores the perspective of senior undergraduate nursing students on mental health and the stigma around it, by considering a simulated case vignette of a person with a mental health problem. A descriptive qualitative approach was chosen and involved three online focus group discussions. The findings show various manifestations of stigma, both at an individual and collective level, which indicates that it is an obstacle to the wellbeing of people with mental illness. Individual manifestations of stigma concern its effect on the person with mental illness, while at the collective level they concern the family or society in general. Stigma is a multifactorial, multidimensional, and complex concept, in terms of identifying and fighting it. Thus, the strategies identified involve multiple approaches at the individual level, aimed at the patient and family, namely through education/training, communication, and relationship strategies. At the collective level, to intervene with the general population and specific groups, such as groups of young people, strategies suggested include education/training, use of the media, and contact with people with mental disorders as ways to fight stigma.

## 1. Introduction

Globally, mental health is a leading cause of ill health and is explicitly prioritized in the 2030 Agenda-Sustainable developmental goals [1]. The importance that mental health has in peoples’ lives is clear. Despite the progress that has been made, mental health patients are still a target of severe rights violations, discrimination, and stigma [2].

Discrimination has been seen as a behavioral response caused by negative attitudes [3]. Various forms of discrimination occur and materialize in differential treatment for certain individuals because they belong to specific groups [4].

Prejudice generally refers to attitudes, emotions, or behavior towards group members, which implies, directly or indirectly, some negativity or antipathy towards that group [5]. Prejudice adds emotional content to affective content (stereotypes), giving rise to an unfavorable attitude [6].

Stereotypes reflect preconceived opinions and attitudes about members of certain groups (e.g., ethnic, religious, mentally ill). The general population has negative and stigmatizing attitudes toward mental health patients, and many myths and stereotypes remain [7,8,9]. Even though the stereotypes differ according to the disease, danger, unpredictability, and guilt are common and frequently result in negative attitudes and discrimination toward people with mental health problems [10,11].

Stigma is distinguished from discrimination, prejudice, and stereotypes; it is described in three conceptual domains (i.e., cognitive, emotional, and behavioral) [12], which can have detrimental effects on the health and wellbeing of people with mental illness [13]. There are several manifestations resulting from stigma, whether individual or collective. Stigma as a threat to one’s identity occurs when an individual evaluates the demands imposed by a stressor of a stigmatizing nature as potentially threatening to their social identity and wellbeing [14]. Labeling and identity theories explain the development and maintenance of stigmatized identities. The person with mental illness loses their previous or desired identity(s) and starts to adopt stigmatizing views of themselves, based on their condition, as useless and incompetent. Thus, the impact of the diagnosis on these people’s identity of ‘being mentally ill’ can become dominant in the lives of some individuals [14].

Stigma is defined by Goffman as an “attribute that is deeply discrediting, reducing someone “from a whole and unusual person to a tainted discounted one” [15] p.3. Subsequent studies agree with this definition [16] and reinforce that stigma is firmly rooted in cultural and contextual issues that differ over time and across contexts [17]. Over time, the definition of stigma has evolved as a product of socialization. Thus, two types of stigma have been underlined: public stigma and self-stigma [18].

The term “public stigma” refers to the stigmatizing perceptions that the general public holds of those with mental illnesses. The internalization of public stigma constitutes self-stigma [18], when people with mental illness internalize public attitudes and as a result suffer low self-esteem and low self-efficacy, compromising recovery potential [12]. Internalized stigma can impede the effectiveness of treatment and the recovery process, as people with mental illness experience feelings of shame and self-devaluation and subsequently withdraw from social activities [19]. Family members of the person with mental illness are also affected by internalized stigma; they feel ashamed and blamed for their relative’s condition, which can lead to isolation and economic difficulties [12].

The stigma attached to mental health conditions is universal, permeating all cultures and contexts in countries worldwide [20]. Only seven percent of people in developed countries think it is possible to overcome mental illness and believe that people with mental illness are more violent. Around 15% to 16% of people in developing countries believe the same [21].

For many people with mental health problems, stigma and discrimination create severe barriers to receiving quality health care [22]. It is expected that, because of their training, the healthcare professional (HCP) has a more realistic and reflexive perspective on mental illness and mental patients, but it is known that stigma also informs attitudes of health professionals [23,24]. Stigma among health professionals is very similar to that reflected by the general public, but it creates an ethical problem regarding the barriers associated with health care [25].

Some studies reveal positive attitudes in nurses [26,27], but others disclose negative attitudes and behaviors among these professionals [24,28], revealing themselves as less optimistic about the prognostic and long-term health of those with mental illness compared with the general public [24]. The experience of stigma in regard to mental illness was similar between nurses’ and patients’ perspectives, namely in terms of time constraints, emotional distress, diagnostic overshadowing, and labeling. In addition, stigma experiences can be organized into three levels: individual, institutional, and social [22]. Nursing students have more anxiety in the presence of mental patients and less knowledge about mental health than nurses; however, nurses have more stigmatizing attitudes [29]. In nursing students, mental health stigma appears to be higher in help, pity, coercion, and avoidance dimensions. It also seems to differ in relation to their year of training, relationships with mental health patients, mental health diagnoses in themselves, and interest in the area [30].

Recent literature reviews highlight the need for professionals to be aware of their stigmatizing attitudes and discriminatory practices to minimize the negative impact on the people they take care of [28,31]. Only a tiny percentage of the research on mental health stigma aims to measure the efficacy of interventions to reduce it, and generally, they occur in small contexts. Presently, the more effective and promising interventions are those that involve persons with living experiences of mental disease [32].

It has been noticed that stigma does not diminish significantly with theoretical learning, and there is a need to input contact with patients into clinical practice [30,33]. Anxiety in facing mental patients, and consequently mental health stigma, was reduced in a group of nursing students after an intervention during their clinical practice in a psychiatric unit. The impact of contact with patients was evident [33,34]. There is a need for more research that targets the impact of stigma reduction interventions on patients’ and in HCP experiences, evaluating this impact on actual actions rather than on behavioral intentions. Additionally, there is a need to assess the effects of interventions with a longitudinal perspective and to use mixed methods that track the implementation of more sustained interventions among health care professionals [34] and students [25,33].

While various measurement tools quantify mental health stigma, we believe there is still a lack of a holistic, comprehensive approach to this issue in nursing students, which is crucial to designing the strategies to curtail barriers to mental health practice, promote help-seeking behaviors, and expand mental health nursing education. To address this gap, this study’s primary goal was to understand senior undergraduate nursing students’ perspectives regarding mental health stigma and discrimination. For this purpose, the study was intended to understand the views of a group of nursing students about mental health, starting with a simulated case vignette of a person with a mental health problem.

## 2. Materials and Methods

### 2.1. Study Design

A descriptive qualitative approach was chosen, which involved online Focus Groups (FG); it was designed using Krueger and Casey’s methodological framework [35]. The Consolidated criteria for Reporting Qualitative research (COREQ) checklist were followed [36].

### 2.2. Sample Selection and Recruitment

Three FG were run, with 19 nursing students (9 in the first group, 6 in the second, and 4 in the third FG) from 38 potential students in the 4th year of the bachelor’s degree program at one Portuguese nursing school. It is generally accepted that between four and twelve participants are sufficient for each FG [35]. The participants were divided according to availability, considering that some were working students. These students had previously, in the third year of training, a curriculum unit about mental health and psychiatric nursing where stigma and discrimination were a topic, and an internship in mental health and psychiatric nursing, where they developed their skills in psychiatric units, in contact with mental health patients, for 9 weeks (250 h of contact on clinical placement).

Participants were selected using a purposeful sampling procedure based on previous clinical experience with a person with a mental illness. The inclusion criteria were as follows: (1) adult nursing students (≥18 years old); (2) can communicate in Portuguese; and (3) agree to participate in this study. We excluded participants who had no access to any electronic device for joining the online interview.

The option of conducting online FG was chosen because there were no classes during the study period, and we wished to ensure low dropout rates. In addition, the threat imposed by the COVID-19 pandemic further strengthened the rationale for conducting FG research online [37].

### 2.3. Data Collection

The online FG sessions took place over two months (October 2022 and November 2022) using the videoconferencing application Zoom, and each session was approximately 70 min long. Before each FG, participants completed a short questionnaire with some sociodemographic and health-related variables. Although they did so from the beginning of the meeting, participants were asked to keep the cameras on, and they complied. During the data collection, all participants had the video cameras and audio-activated to ensure the visualization of the nonverbal cues in their interactions. This information was negotiated before each focus group started. The FG sessions were recorded with the participants’ permission and then transcribed for a more efficient analysis of the discussion content and the answers.

The sessions started with a simulated case vignette of a victim of stigmatizing attitudes toward mental health: John is a 22-year-old university student with anxiety symptoms. He had several anxiety crises and panic attacks, which his colleagues interpreted as a way to get an advantage in school evaluations. They did not understand why he behaved this way, because they judged that he should be able to control the symptoms. After the case presentation, a discussion was held with a group of questions (Table 1), and participants were encouraged to give their opinions. Each FG was conducted to explore participant opinions about stigma in the lives of people with mental illness. The data were collected using a semi-structured interview guide with open questions, which allowed us to get to know the participants’ points of view better, and to get a more explicit idea of their perceptions and experiences. The interview guide was developed based on previous literature as well as the authors’ practical expertise in the research topic [38].

### 2.4. Ethics

The study was conducted following the Declaration of Helsinki and approved by the Local Ethics Committee (protocol code P06-2022, approved on 19 September 2022). Written informed consent was obtained from each participant before participating in an FG. The moderator thoroughly explained the study’s aim and the participants’ right to confidentiality and anonymity. Videoconferencing software Zoom allowed for encrypting all FG sessions and for using a passcode to protect the privacy of a focus group meeting.

### 2.5. Data Analysis

The content of the answers was analyzed using Braun and Clarke’s methodology of reflexive thematic analysis [39]. This approach was justified because it allowed a deductive “top-down” method or an inductive ‘bottom-up’ method to explain themes, highlight similarities and differences, and generate unanticipated insights [39]. This option was in line with the objective of the current study. All the FG interviews were transcribed and imported into MAXQDA qualitative software program (VERBI Software 2021, Berlin, Germany) for data management.

According to Braun and Clark [39], the thematic analysis includes six steps. The first is familiarization with the data by reading all the transcripts. The second step concerns the generation of codes. Reading the transcripts allowed the relevant information to be grouped into intangible codes.

In the third stage, which corresponds to the search for topics, the codes were categorized into topics likely to be related to stigma. The review of themes (fourth step) was confirmed by reviewing all codes and the dataset.

Fifth, topics were defined and named, producing a final thematic tree. Finally (sixth step) a thematic review was used to help write the report. Coding involved two (co)investigators: the principal investigator (O.V.) coded all transcripts, which were then coded independently by co-researchers with scientific training and competence in data processing (T.C. and L.M.). When differences or disagreements arose during coding, the coders conferred until a consensus was reached. A fourth researcher (C.T.) was consulted to reach a final decision. Interrater reliability was calculated using the percentage of agreement [40]. Thus, we considered a percentage value equal to or greater than 90% to represent a consensus.

Selected quotes from the Portuguese transcript have been translated, in italics, and inserted in the following report. Example extracts from the participants were numbered according to the FG number (1–3), by sex (XX/XY), and by interviewee number.

### 2.6. Study Rigour

The interviews were planned and conducted by the research team consisting of the principal investigator and the co-researchers who had insiders’ points of view and knowledge about stigma in the person with mental illness. To ensure reliability and validity, the interviewers underwent training and reflection sessions, including a discussion on the FC script. During the data collection period, the FC facilitators participated in regular feedback sessions based on the recorded FC.

The principal investigator (professor of mental health nursing) was effective in conducting engaging conversations with participants; this facilitated getting to know the participants and having empathetic and assertive relationships. The principal investigator showed neutrality, without value judgments, providing some time for the participants to expose their doubts, difficulties and to make some comments about their experiences in an internship in mental health and psychiatric nursing.

The findings were reviewed by co-authors (researcher triangulation) and resulted in the revision of theme and sub-theme names to legitimately capture the essence of the findings. We also maintained an audit trail, which included field notes, coded transcripts, and comments and reviews from group coding meetings, following best practices in qualitative research.

### 2.7. Research Team and Reflexivity

The research team consisted of nine certified mental health nurses and one certified medical-surgical nurse-all with experience in conducting studies using qualitative approaches. The principal investigator (Ph.D.) and two co-investigators (Ph.D. and doctoral student) followed the FG interview. All were specialists in Mental Health and Psychiatric Nursing in a School of Nursing. They had previous research experience and were familiar with the student group. One of the co-investigators was the secretary of the FG sessions.

The students were informed about the researchers, their personal goals, and the reason for the research.

## 3. Results

### 3.1. Sample Description

The 19 participants were mainly females, had no previous diagnosis of mental illness, and evaluated their mental health in most cases as good (*n* = 7) and very good (*n* = 5) (Table 2). Nonetheless, some respondents reported mental health symptoms such as anxiety and depression. Most of the students had a friend or relative with a mental illness.

### 3.2. Themes and Sub-Themes Emerged

From the data analysis, done with a reflexive thematic analysis (Table 3), emerged three themes and five subthemes, with 14 different codes.

The data relating to the attributed meanings of stigma were analyzed according to the meaning of social stigma [41], which has three components: stereotypes, prejudice, and discrimination.

Manifestations of stigma emerged, whether at an individual or collective level, which indicated that it was an obstacle to the wellbeing of people with mental illness. Individual manifestations concerned the person with mental illness, while at the collective level, they referred to the family or society in general.

The strategies for stigma reduction were grouped according to their individual or collective nature. Individual strategies referred to strategies directed to the person with mental illness and/or their family, while collective strategies referred to strategies directed toward groups or society in general. In the following subsections, the findings presented in Table 3, are given by sub-themes and codes.

#### 3.2.1. Individual/Collective Attributes


Stereotypes


The participants identified rigid ideas and attitudes involving negative and harmful emotional content for the person with mental illness, with expressions appearing that labeled them as “*mad*” (FG1-XX-4), “*crazy*” (FG2-XX-3), “*they start calling him crazy, crazy boy and say that they have no patience for crazy people*” (FG1-XX-2).

Other situations were identified in which the participants believed that the behavior presented by the person with mental illness was aimed to obtain gains “*he’s making a big show... as a way of him to get some advantage*” (FG3-XY-1), not understanding the reason why it was happening.

The behavior of health care professionals regarding the existence of stereotypes towards people with mental illness was highlighted by the participants when they stated that
“*because he’s crazy, he doesn’t deserve it, I do not have patience (...)*” (FG2-XX-2), “*When I started my internship, I had a patient with schizophrenia and I couldn’t get to her, (...) I realized that the problem was me, not her because I was afraid of her, I was afraid that she would attack me, I don’t know... I was afraid of not speaking the right words to her, of saying something that would trigger her madness, her irritation, and her aggressiveness*” (FG3-XY-3); “*When I went to the internship, I was afraid that I wouldn’t be able to interact with people and would have some communication difficulty*” (FG1–XX-5). One of the participants gave the personal example of “*a relative who has a mental illness that went to an emergency room and because he was mentally ill, we get a green bracelet, but he was having a stroke*” (FG3-XY-1), “*I sat down at the coffee table and one man came up to me and started asking me for help messing with my phone and stuff like that, and I really got to thinking, what does this crazy guy want*” (FG2-XX-4).


Prejudice


When talking about the meanings of stigma, participants referred to prejudice in expressions such as
“*... the fact that the young man had an anxiety attack during the exam and they immediately thought that what he was going through was not exactly physiological, it was unclassifiable*” (FG1-XY-1), “*... if it was another disease like a stroke, for example, that had these symptoms, maybe they would pay more attention*” (FG2-XX-6), and “*an example (...) in my family, a teenager asked his mother to see a psychologist because he felt stressed and had panic attacks and his mother said that he had to go to school because that was an adolescent problem and would pass*” (FG3-XY-1).


Discrimination


The behaviors described by the participants as discriminating were criticism
“*colleagues criticized him and kept away from him*” (FG2-XX-6), “*when they criticize him and do not try to understand his colleague’s situation*” (FG3-XY-1), discrediting the behavior of the person experiencing stigma, non-verbal communication “*non-verbal communication, the way they look at him, I think that’s where stigma starts*” (FG2-XY-1), devaluation “*it’s related to what colleagues said ... with the devaluation of him*” (FG1-XX-8), “*I have a relative who has a mental illness and went to an emergency room and as he had a mental illness and was a nutter the green bracelet was put on and he was having a stroke, so the symptoms were devalued*” (FG3-XX-2) and the discredit towards those with mental health problems “*if someone knows that he is undergoing treatment for something ... the person might not be seen with that much seriousness and confidence*” (FG2-XY-5) “*Knowing that a colleague has a mental illness will generate negative thoughts and negative feelings*”(FG1-XX-5); the fact of using therapy to deal with situations related to shame and difficulty in talking about the problem was also mentioned “*we once had a patient who was homosexual and because he was homosexual and did not want to assume it and had difficulty to say it to his family, they gave him an antidepressant, they always gave him only medication, but the problem was something else, it was losing his shyness and say that he was homosexual and not hiding through medication*” (FG1-XX-3).

#### 3.2.2. Manifestations of Stigma at the Individual Level

Students identified manifestations of the stigma that could threaten the identity of the stigmatized person, and the way they feel: “*…a stigmatized person may feel fear or shame, which can lead to anxiety and depression*” (FG1-XX-2). The stigma according to the participants can make them feel that they are not normal, that they have no value, with a decrease in self-esteem and self-confidence: “*…the person feeling that he is… different, that he is crazy, something that is not it’s normal*” (FG1-XX-6). The person may still believe “*that he or she is good for nothing, therefore, self-esteem is greatly affected*” (FG2-XX-3); “*many situations and attitudes that people have, make us believe that we are disabled*” (FG3-XY-1). These beliefs can affect the person’s life: “*the feeling of inferiority concerning other people… it leaves scars for the rest of their lives*” (FG3-XX-2).

At the individual level, they also mentioned self-stigma/internalized stigma, that is, they referred to a perception of devaluation, marginalization, shame, and isolation:
“*in addition to the person who is dealing with the stigma of others, he may eventually come to deal with his stigma, self-esteem, that is, from dealing with these situations so much he ends up realizing that there is something wrong with him and that he has to change, that he is not well*” (FG1-XX-3); “*…the stigma, … has to do with labeling, with an exclusion, which is often done concerning a situation, a specific person, which can even lead to discrimination itself*” (FG3-XY-1).

Feelings of shame and derogatory feelings are mentioned: “*…people who suffer from mental illness are ashamed, they are ashamed…*” (FG2-XX-6); “*a look in a strange way, you can see it as being judgmental or recriminating him, taking away his self-confidence, causing him to have derogatory feelings about himself*”. Related to the internalized stigma, people withdraw from social activities: “*…the person does not feel understood, feels isolated or isolates himself.*” (FG3-XX-2).

Inability/resistance to seeking help was also unanimously mentioned: “*instead of asking for help and getting better, it turns out to be the opposite*” (FG1-XX-5). Stigma can affect medication adherence:
“*…the responses he received from society, instead of helping him progress in dealing with the disease, made it even more difficult to the point where he wanted to hide it and not want to take the medication…, that is, he did not adhere to the therapy because he felt this stigma, because he felt he was not like the others*” (FG2-XX-6).

Indeed, the stigma may lead the person not to accept their (mental) illness and resist seeking treatment: “...*the denial on the part of the person going through the situation and resistance to not seeking treatment*” (FG3-XY-E1).

#### 3.2.3. Manifestations of Stigma at the Collective Level

The participants identified manifestations of stigma in terms of social rejection: “*social exclusion often happens in face of mental illness, …*” (FG1-XX-4); “*…what happens is exactly the distancing from society, rejection…*” (FG2-XY-5). The stigma of mental illness can be so harmful that it leads family members not to accept the illness of that family member—changed familiar dynamics—“*...the family itself sometimes does not accept that the family member has to take psychiatric medication*” (FG1-XX-6); “*sometimes families try to hide it…*” (FG2-XX-3). Other family members isolate themselves: “*many families isolate themselves or try to hide it*” (FG-3XY-1).

Some myths associated with mental illness were also identified in the students, such as, that people are dangerous: “... *I was afraid of not saying the right words to her (person with mental illness), of saying something that triggered her madness, her irritation, and her aggressiveness … I was afraid of her; I was afraid that she would attack me*” (FG3-XX-2). Some of the participants mentioned their stigma: “*ourselves, as we said in class, suffer from stigma*” (FG2-XX-6).

#### 3.2.4. Individual Strategies to Reduce Stigma

The participants identified education and training strategies for the person with mental illness and his/her family:
“*... I agree with health education. I think that primary health care units could play a more interventive role because they end up receiving a large part of these family members in the appointments. They also can bring together, for example, the children and the parents, i.e., they could take advantage of it and work on this topic with the family members. I think that it would also be important to create moments in which people who have already suffered stigma or who suffer stigma could meet and talk*” (FG2-XX-3).

At the individual level, they also mentioned communication and relationship strategies with people with mental illness:
“*... it is important to understand, to know how to maintain calm, to know how to talk to people, sometimes even lowering ourselves to their level and trying to understand what the problem is, what is going on, why they are feeling this way and what can I do to help them*” (FG1-XX-2).

This strategy was mentioned with implementation by health professionals (“*many of the users of the institution say that they would like all professionals to have a little time to listen to them... and not to treat them differently*...” (FG1-XX-3)), but also by society in general
(“*... if we were more human, I think that’s the right word to use, more understanding with others, we would pay more attention to these things, without that hatred, to worsen the situation of the person who already has a certain difficulty in dealing with some things. This difficulty can be overcome with the support of someone close, a colleague from college, a neighbor, or all, ....*” (FG3-XX-5).

#### 3.2.5. Collective Strategies to Reduce Stigma

Within the collective strategies, it was also possible to identify education and training strategies:
“*I feel that this is so ingrained in our culture, I don’t know if it’s cultural or if it has to do with society, but I think that there is already so much information and yet this type of behavior persists. I think that there should be more dissemination, more information for us to understand that this is something that can happen to anyone, that it is an illness, it is a mental illness and should not be stigmatized*” (FG1-XY-9).

This intervention may occur at a societal level or in groups:
“*I would like to say here something that I think would be important, which is the organization of more lectures in schools, such as lectures on sexually transmitted diseases. I never attended one about mental illness and I think that education also starts, mainly, with the younger people and I think that it would be important to invest more in this...*” (FG1-XX-4).

Strategies involving the media to make mental health visible: “... *to work on the sensitization of journalism and the media, i.e., to provide more balanced information*...” (FG3-XY-1) and “... *avoid perhaps sensationalist types and language in the media*...” (FG2-XY-5). In particular, the use of television news, soap operas, or reports were evident in the group’s discourse: “*I think, for example, that soap operas are great disseminators of knowledge on this issue*” (FG2-XX-6).

The sharing of experiences by the person with mental illness and, to the same extent, the contact with people with mental illness were also mentioned: “*the stigma is in all of us, not in all of us, but in most of us and I think that our stigma, I think I can speak for all of us here, has greatly reduced after the internship and after we have contact with people with mental illness*”(FG3-XX-2).

## 4. Discussion

Even when the morbidity and mortality of mental illness are low, but the condition is highly stigmatized, the burden of stigma can exceed the burden of the illness, namely impacting social, emotional, and professional functioning, thus negatively affecting the person’s quality of life [42]. Stigma has long been recognized as a significant challenge in treating and recovering from mental illness [43], for which several individual and collective attributes compete. The use of a case vignette can be considered an added value because it facilitates the evaluation, allowing responses from nursing students about stigma in mental health, because they felt closer to a real situation.

Stereotypes represent cognitive schemes that contain beliefs and opinions about the characteristics, attributes, and behaviors of the members of various groups [44], such as people with mental illness. Regarding the presented vignette, the person with mental illness is characterized by such expressions as “*crazy*” or “*nutty*”, that is, labeling, which came from the recognition of differences or marks in the character’s behavior [45]; it was inferred that those behaviors are those of a person with mental illness without any evidence that this is true [46].

Stereotypes of the person with mental illness bring disadvantages to themselves and their participation in life in society [47]. In the vignette presented, stereotypes of the boy’s colleagues were evidenced in the way they evaluated his behavior in the anxiety situation, as they believed it was aimed at gaining advantages. People with mental illness are portrayed in literature as being incompetent, dangerous, childish, and unable to take care of themselves [48,49], using their mental health problems as a way of obtaining gains. This can be explained by misinformation about mental illness [50].

The existence of stereotypes about mental illness is identified in the general population but also among health professionals. They are not safeguarded from internalizing the stereotypes assumed by society [51] which may be reflected in their attitudes and behaviors. The existence of expressions such as “*When I got there in the internship, I had a patient with schizophrenia, and I couldn’t get to her*” shows the existence of stereotypes on the part of health professionals. The same author points out that the higher level of training of professionals reduces attitudes and behaviors related to stigma.

Evidence shows that stereotypes are evident in general society and among health professionals [52]. In addition to those already referenced in the literature, our findings highlight that the act of assuming that the person adopts behaviors compatible with those of a mental illness for gain is also a stereotype.

Attitudes that imply some negativity may be the evaluation of behaviors such as “*those actions the boy has in the face of anxiety something unqualified*” and devaluation about mental illness problems demonstrated in expressions such as: “*.... if it was another disease like a stroke, for example, that had those symptoms, maybe they would pay more attention*” which is confirmed by some authors [53,54] when they state that people with a mental disorder are less likely to be observed and assessed by specialists when they complain, which results in premature deaths.

Prejudice is also expressed in the difficulty in accepting that a family member may have a mental health problem, as can be seen in the expression “*because he felt stressed and had panic attacks and his mother said that he had to go to school, because that was a problem of adolescence, and it would pass*”. Not accepting the existence of the problem may be related to the belief that those suffering from mental illness are not able to recover from their condition and will never be able to work or live independently [53].

The stereotypes identified by the participants seem to materialize in prejudices such as devaluation both in the care provided by health professionals and in family and social interaction.

The discrimination behaviors identified, including pushing away “*colleagues (...) would push him away*” are corroborated by literature [53]. The literature states that the stereotype that a person with mental illness may pose a danger by their aggressiveness leads to rejection behaviors [55]; however, the participants talk about pushing away without having identified aggressive or dangerous behaviors. The situation portrayed in the case vignette could explain the detachment from peers and the unpredictability of the character’s behavior [47].

In addition to disengagement, other discriminatory behaviors were identified by participants as the fact that the person is not viewed as “*serious or trustworthy*”, which may have consequences in terms of finding a job, sharing accommodations, or even starting an affective relationship [54,56]. The discredit assigned to people with mental illness may explain the high unemployment among people with mental illness [57].

Aware that they may be stigmatized, people with mental illness sometimes adopt behaviors such as “*hiding the therapy*” [58], which may harm treatment adherence [59].

Discrimination, according to the participants, can manifest itself in the form of verbal and non-verbal behaviors, i.e., in various forms of communication and interaction. Communication is implicitly necessary to achieve mental health, build social relationships, and be an active participant in the world [60]. However, stigma includes discrimination, and discrimination, in general, undermines the building of social relationships. It seems therefore important to reflect on the different forms of communication with the person with mental illness as they may contribute to their stigmatization.

Our findings show the existence of discriminatory attitudes in the distancing, discredit, and lack of trust toward people with mental illness. Non-stigmatized people try to avoid interactions with stigmatized people, which contributes to a decrease in the size and quality of the social networks of stigmatized people.

In this sense, the participants were unanimous regarding the experience of stigmatization affecting the identity of those with mental illness, leaving scars, and resulting in loss of self-esteem, attributing to themselves characteristics of incompetency [61].

Loss of identity seems to be a major factor in the experience of internalizing stigma/self-stigma. The literature shows the negative effect of stigma and discrimination and its role in shaping the internalization of stigma [62,63]. Society’s perceptions influence beliefs about how people with mental illness see themselves, that is, they develop negative stereotypes and prejudices towards them and exhibit secrecy and isolation behaviors to deal with this discrimination. These results are in line with the present study, in which internalized stigma was related to beliefs of self-devaluation and self-discrimination with decreased self-esteem, self-efficacy, and worsening of symptoms [14,64,65].

The fear of being stigmatized or labeled is one of the main reasons why many people living with mental illness do not seek help. The stigma associated with mental illness prevents people from making use of community resources [66]. Many people with a stigmatized health condition do not seek help, which delays diagnosis and treatment. Also, in the present study, resistance to seeking help is mentioned, which may aggravate the mental illness and reduce the likelihood of recovery. In this sense, adherence to treatment for mental illness becomes a challenge for HCP [67].

Social rejection was also identified by students as a collective manifestation of stigma. These results are in line with other studies that suggest that rejection experiences were more prevalent among people with mental illness [68,69].

Stigma not only directly affects individuals with mental illness, but also the loved ones who support them, often including their family members [70]. It should be noted that the participants in this study reported that the family has difficulty understanding the mental illness and may deny or hide the condition. Research shows this inability to accept the disease by family members [71].

With a lower frequency, some myths/beliefs associated with mental illness were addressed by students, such as the person with mental illness being aggressive and showing stigmatizing and fearful attitudes. These results are in line with a study carried out with medical and psychology students [72].

Therefore, there is an urgent need for specific intervention strategies to combat stigma. In this sense, participants were unanimous about the need to develop strategies to reduce stigma. In the thematic analysis of the data, it was possible to identify strategies that were grouped according to their collective or individual nature. These findings agree with the available evidence that identifies stigma as a global phenomenon, which requires intervention approaches that target multiple levels, including the individual, interpersonal, community, and structural levels [73].

In the realm of individual strategies, it was mentioned that education/training of the person with mental illness and/or family is an important strategy in fighting stigma. This strategy is often identified in the literature as a psychoeducational intervention with the potential to help patients gain more knowledge about mental illness and improve their understanding and ability to cope with the stigma of mental illness [74,75]. Such interventions mainly contribute to critical thinking about understanding of mental illness but does not reduce the perception of stigma and internalization [76]. Other studies also confirm that the use of the psychoeducational strategy alone was limited in improving self-stigma [74]. On the other hand, although it was not mentioned by participants, still in the context of education/training for people with mental illness, evidence shows that peer-led intervention can reduce self-stigma and stigma pressure, improve recovery and empowerment, and increase self-efficacy and willingness to seek professional help [77].

Family education/training is also identified as a possible strategy in other studies that point to positive results such as changes in affective attitudes, cognition, and behavior, namely, family members learn to act, express their feelings, and think in a flexible way about the disease, but there is no significant evidence on the results of these interventions on stigma [78].

The other strategy at the individual level is related to communication and relational strategies with the person with mental illness, both by health professionals and society in general. Communication is essential in any context, particularly in health, but it is even more important in mental health, due to both the nature of the problems and their potential impact [79]. This intervention refers to the development of skills at this level. There is no evidence available on this strategy as a resource to combat stigma, which is why we consider it to be an innovative finding in this study. Thus, interventions for the development of communication and relational skills should be part of anti-stigma intervention programs.

At the collective level, the education/training of society or groups was also mentioned as a strategy to be adopted in the fight against stigma. About the education of students on this topic, referred to by several participants, a systematic review of randomized controlled trials for the identification of anti-stigma interventions on mental health targeting students identified that education through lectures and case scenarios, contact-based interventions, and role-playing as strategies to combat mental illness stigma demonstrated significant improvement in attitude towards mental illness and help-seeking. Mental health knowledge also helped to reduce both public stigma and self-stigma and improved literacy and attitude towards mental health illnesses [30,34,80]. As mentioned in this review, participants also referred to sharing experiences and contact with service users, namely in their internship context, as effective in combating stigma. Another systematic review on attitudes and stigma toward mental health in nursing students reinforces that clinical placements in the field of mental health are essential for promoting positive changes in student nurses’ attitudes and stigma [33].

In the domain of collective strategies, the media were pointed to as a way to uncover stigma through programs, journals, and reports. The available evidence indicates that persuasive and purposeful education aimed at the public to correct misconceptions associated with mental illness, with attention to language, can help reduce stigma in mental health [81].

Other strategies are identified in the literature but not in this study, like protest and advocacy, policy, and legislation changes [82]. Possible reasoning for this fact is that the participants are nursing students, with training more focused on direct interventions with people. Even so, this fact should be assessed and reflected upon so that future health professionals have a more comprehensive view of health in general and mental health in particular. According to the participants and the currently available evidence, the strategies to be implemented can be complementary and not implemented in isolation.

### 4.1. Study Strengths and Limitations

A strength of this study is that it takes the in-person perspective and explores reflection-beyond-action. The interaction between participants, which is a feature of FG, stimulates thoughts and allows respondents to add their experiences and sentiments that would not be expressed in individual interviews, and this appears to have been accomplished. The option of conducting online FG proved to be an appropriate option with a high level of engagement, perhaps because participants felt comfortable and safe at home during their participation. Furthermore, young people become a “digital native” generation, developing digital competencies by attending online classes and virtual social networking from home [83].

By giving nursing students a voice and anticipating self-analysis and self-reflection in the face of stigma in mental illness, this study contributes to making them aware of their internalized stigma, which becomes central to intervening responsibly and therapeutically.

This study also has some limitations, including the sample size and the fact that the sample population only included fourth-year nursing students from only one institution. It should be replicated with participants from other academic backgrounds and different levels of expertise and competence. Moreover, some constraints of performing online FG were mentioned, such as difficulty observing body language, missed non-verbal cues, and the distractor environment in some cases.

### 4.2. Study Implications

This study promotes the training of nurses in general, particularly mental health and psychiatric nurses, as well as teachers, for the recognition and effective and efficient action involving individuals/families with mental illness and who suffer from stigma and discrimination, easing the difficulties surrounding these feelings. Another implication related to education is the need to reorient nursing curricula through critical reasoning skills and competency-based education [84].

On the other hand, the implementation of research activities, in the context of stigma, involving students, can be a way of learning through experience, in a collaborative process and joint development of investigative and reflective skills that can contribute to the change of school culture and, ultimately, for social change.

## 5. Conclusions

Based on our findings, mental health stigma can make the person the target of prejudice, isolate themselves, be reluctant to seek help, and consequently, possibly not adhere to treatment. In this sense, the perception of nursing students about stigma and its manifestations is necessary to reflect on the strategies to be adopted to reduce stigma. Stigma is a multifactorial, multidimensional, and complex concept, in terms of identifying and fighting it. The strategies identified refer to multiple approaches at the individual level, aimed at the patient and family, namely through education/training and communication and relationship strategies—and at the collective level, in order to intervene with the general population and specific groups, as is the case of young groups with strategies such as education/training, use of the media as a way to fight stigma, and contact with people with mental disorders. These interventions had already been identified with evidence in other studies, except for communication and relationship strategies, so we suggest that interventions for the development of communication and relationship skills should be part of anti-stigma intervention programs, as well as further studies in this area.

## Figures and Tables

**Table 1 ijerph-20-04601-t001:** Questions of the FG guide interview.

	Questions
1.	Based on the vignette, what does stigma mean to you?
2.	How do you characterize stigma? (or, what are the characteristics that you identify in a person with stigma?)
3.	What are the main consequences of stigma?
4.	What can you do as a future health professional to fight stigma? or, in your case, as a group, what do you think you could do to fight stigma?
5.	Is there anything else you would like, or think is important to share?

**Table 2 ijerph-20-04601-t002:** Demographic characteristics of the sample.

Variables		*n*		
		FG1	FG2	FG3
Sex	Male	3	2	1
	Female	6	4	3
Age range (years)	20–25	4	3	2
	26–30	1	1	1
	31–40	2	1	1
	41–50	2	1	-
Nationality	Portuguese	7	4	3
	Brazilian	1	2	-
	Portuguese/Brazilian	1	-	-
	Rwandan	-	-	1
Working-student	Yes	4	2	3
	No	5	4	1
Marital status	Single	7	4	3
	Married/Consensual union	2	2	1
Presence of mental health symptoms	Yes ^1^	2	1	1
	No	7	5	3
Having a friend or a relative with a mental illness	Yes	6	4	3
	No	3	2	1
Self-evaluation of mental health	Very bad	1	-	-
	Bad	1	1	-
	Regular	2	1	1
	Good	4	2	1
	Very good	1	2	2

^1^ Anxiety or Depression.

**Table 3 ijerph-20-04601-t003:** Themes, sub-themes and codes emerged obtained from the thematic analysis.

Theme	Sub-Theme	Codes
Attributed meanings to stigma	Individual and Collective Attributes	• Stereotypes
		• Prejudice
		• Discrimination
Stigma manifestation/effects of stigma	Manifestations at the Individual Level	• Threat to the identity of a stigmatized person
		• Self-stigma/Internalized stigma
		• Inability/Resistance in searching for help
	Manifestations at the collective level	• Social rejection
		• Changed familiar dynamics
		• Myths
Strategies to reduce stigma	Individual Strategies	• Training/Education
		• Communicational/relational strategies
	Collective strategies	• Training/Education
		• Media/Unmasking the stigma (TV programs, newspapers, reports)
		• Sharing experiences/contact with patients

## Data Availability

All data generated or analysed during this study are included in this article.

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
