# Peer review of "“Looking beyond Mental Health Stigma”: An Online Focus Group Study among Senior Undergraduate Nursing Students"

_ijerph, 2023, doi:10.3390/ijerph20054601_

Round 1

Reviewer 1 Report

As the authors rightly state, conducting research activities, in the context of stigma, involving students can be a form of experiential learning that can contribute to changing school culture and ultimately to social change. In my opinion, in today's society, such studies are very much needed and I am very grateful that I was able to participate in the review.

The way of expressing myself and the methodology were appropriate. I would just like to make two points for your consideration:

The researchers have selected 3 focus groups and explained the number of participants in each group as well as the recruitment method (purposive sampling). However, they do not explain why they have been grouped in such a way. Could they explain the criteria or whether it was done randomly? Lines 107-117. Considering that only final year students were included, explaining this criterion would strengthen the methodology.

Secondly, table 3, although it expresses the themes and sub-themes, is not easy to read. I am not sure if it is the central alignment of the text, the excess of lines or the size of the text. I recommend aligning the subtopics differently and changing the font size or making a diagram.  

Thirdly, the conclusion is too long. The results section and the discussion section go into sufficient depth. I recommend a summary to speed up the reading.

Author Response

Reply to the reviewer 1 – ijerph-2168122

We thank the reviewer for all comments that allow us to improve the quality of our work. All questions have been carefully analyzed, and we have tried to fully respond to what was requested. All changes were highlighted in grey.

As the authors rightly state, conducting research activities, in the context of stigma, involving students can be a form of experiential learning that can contribute to changing schoolculture and ultimately to social change. In my opinion, in today'ssociety, such studies are very much needed and I am very grateful that I was able to participate in the review.

Response: Thank you very much for your positive appraisal and comments.

 a) The researchers have selected 3 focus groups and explained the number of participants in each group as well as the recruitment method (purposive sampling). However, they do not explain why they have been grouped in such a way. Could they explain the criteria or whether it was done randomly? Lines 107-117.Considering that only final year students were included, explaining this criterion would strengthen the methodology.

Response: Thanks for pointing out this issue. According to your suggestion, we reorganize the section method with the following response (Lines 145-162):

“Three FG were run, with 19 nursing students (9 in the first group, 6 in the second, and 4 in the third FG) from 38 potential students in the 4th year of the bachelor's degree at one Portuguese nursing school. It is generally accepted that between four and twelve participants are sufficient for each FG [35]. The participants were divided according to availability, considering that some were working students. These students, had previously, in the third year of training, a curricula unit about mental health and psychiatric nursing where stigma and discrimination are a learning topic, and an internship in mental health and psychiatric nursing, where they develop their skills in psychiatric units, in contact with mental health patients, for 9 weeks (250 hours of contact on clinical placement).

Participants were selected using a purposeful sampling procedure based on previous clinical experience with a person with a mental illness. The inclusion criteria are as follows: (1) adult nursing students (≥18 years old); (2) can communicate in Portuguese; and (3) agree to participate in this study. We excluded participants who had no access to any electronic device for joining the online interview.

The option of conducting online FG was due because during the period in which they were held, there were no classes, and, in this way, we tried to ensure low dropout rates.  In addition, the threat imposed by the COVID-19 pandemic further strengthened the rationale for conducting FG research online [37].”

b) Secondly, table 3, although it expresses the themes and sub-themes, is not easy to read. I am not sure if it is the central alignment of the text, the excess of lines or the size of the text. I recommend aligning the subtopics differently and changing the font size or making a diagram.

Response: Thank you for pointing this out.  We responded accordingly by reshaping the table, text alignment and front size and revising the lines. Please see the table 3 in page 7 (Line 268).

 c) Thirdly, the conclusion is too long. The results section and the discussion section go into sufficient depth. I recommend a summary to speed up the reading.

Response: Thank you for your pinpoint. As recommended, the conclusion was changed to be more concise (lines 604-617).

We hope we have accomplished all the suggestions to consider this paper to be published. Thank you again for helping us to improve our work.

Yours sincerely

The authors

Reviewer 2 Report

I would like to thank the editors and authors for the opportunity to review the article LOOKING BEYOND MENTAL HEALTH STIGMA: A FOCUS GROUP STUDY AMONG NURSING STUDENTS.

In general terms, I can state that the article presents a good bibliographic base, with recent citations and specific to the area of interest.

As I will comment below, I have the impression that the discussion section contains some basic and relevant information that should be presented in the introductory section. I also note some methodological issues in the qualitative analysis, design, etc... that should be corrected and elaborated with more precision.

The chosen topic is an interesting one of greater importance considering the increase of mental illnesses and disorders in pandemic COVID-19. However, it is a widely treated topic, especially through quantitative studies and larger samples. I do not mean that a qualitative approach is poorly chosen, rather I mean that this type of approach requires that it be accompanied by information that I do not find in the manuscript, and, further justification or representation of the results in FG1, 2 and 3.

I will now offer my contributions or suggestions for improving the text:

TITLE AND ABSTRACT

The focus group is conducted through the online system, therefore, it is important to indicate this condition in the title and abstractà  an online focus group.  Indicate in the abstract that the patient is simulated.

1.- INTRODUCTION

The introduction needs more development, you can move some paragraphs from the discussion section to this section (see my suggestions in discussion).

I am struck by the fact that they only use the definition of stigma proposed by Goffman (1963). I suggest reading:

Rossler, W. The stigma of mental disorders: A millennia-long history of social exclusion and prejudice. EMBO Reports, 2016. 17(9); 1250-1253. https://doi.org/10.15252/embr.201643041

It may help in a synthetic and summarized way to provide a historical overview of the use of the STIGMA concept and the studies that have been conducted. I mean by this that it is important to justify why Goffman's theory is used and not others. Therefore, it is convenient to address the evolution of the concept and reflect it in the article.

The present study is basically an extension of the study presented by one of the co-authors of the present manuscript in 2022: Impact of an intervention on stigma in mental health and intergroup anxiety DOI: 10.37689/acta-ape/2020AO0226

The reason for wanting to understand the phenomenon of stigma qualitatively must be well justified.

In the last paragraph of the introduction (lines 96 to 99), it should be indicated that the research is carried out on a small sample of fourth-year students, and that the stimulus is a simulated patient, not a real one.

2.- MATERIAL AND METHODS.

2.1.- Study design:

They should indicate the motive or reason for not conducting the Focus Group in person,

Sample selection and recruitment:

Indicate the number of students who were in the 4th year, and also specifically the nursing school (Ribeiro Sanchez or others??).

Data Collection:

Indicate if the participants answered with the video image activated, and if there was any protocol to control the attention and participation of the students.

They must clearly indicate that "John" is a simulated patient (line 125).

On line 132 they should indicate the type of interview: structured, semi-structured, etc....

2.5.- Data Analysis

They should indicate the advantages or reasons for choosing Braun and Clarke's model and not that of other authors, e.g. Smith IPA.

https://www.google.com/url?sa=t&rct=j&q=&esrc=s&source=web&cd=&ved=2ahUKEwjXz7GisJL9AhUgXaQEHcEGCNQQFnoECA0QAQ&url=https%3A%2F%2Fmed-fom-familymed-research.sites.olt.ubc.ca%2Ffiles%2F2012%2F03%2FIPA_Smith_Osborne21632.pdf&usg=AOvVaw1svhzlgxwzZXODOdivaV_x

I respect your choice, but considering the objective and content of the manuscript, I think Smith's model is more appropriate. Please indicate why you do not use phenomenological analysis (IPA) in the Focus Group.

In lines 156-157 I miss the Kappa indicators that numerically reflect the level of consensus on the three main themes identified.

3. RESULTS

Sample description

In lines 190 to 192 it is stated that there are no previous diagnoses of mental illness and in table 2 it is stated that there are four cases. The level of mental health is defined as good or very good, and in the table there are 7 cases of very bad to regular levels.

The composition of FG1, FG2 and FG3 is not described in detail. Given how small the sample is, and that it is not reported why the groups are not balanced with the same or similar number of participants, they should make a detailed table with the same variables used in Table 2, but distributing by FG1 FG2 and FG3. This may be interesting to understand the level of depth with which the issues are addressed in the different FGs.

3.2. Themes and sub-themes emerged

Four sub-themes and 14 different codes are mentioned. If I am not mistaken, Table 3 cites 6 sub-themes, and 16 codes.

In the coding of the literal discourse (3.2.1, 3.2.2. and 3.2.3) the same format in italics should be used. Please note that at other times you use only quotation marks without italics. At the end of each discursive quotation, you should indicate the group and the code of the interviewee and sex, for example FG1-XX-1, or FG3-XY-4, etc....

In section 3.2.1., stereotypes, FG2 is underrepresented, there is no expression of these participants. The same in Prejudice, FG1 is omitted, and in section 3.2.4, the opinions of group FG3 are not taken into consideration. The three FGs must be given a voice in some way, this underrepresentation cannot occur, with the problem that the three FGs are unbalanced with different numbers of participants.

4. DISCUSSION

This is a very extensive discussion, which contains elements that could well be moved to the previous sections. Lines 349 and 350 can fit in the procedure section. Lines 378 to 381 (prejudice) can fit in the introduction. Limit the discussion only to commenting on the results with reference to existing results in other studies.

Lines 471 to 474 do not justify the level of consensus or unanimity that the authors refer to.

Study Strengths.

You say that interaction among participants is a strength of the FG, but this is more so when it is approached face-to-face. Here the format is online, and this entails some very important limitations that the authors do not indicate. It should be clearly stated what the problem with online FGs is.

FINAL DECISSION:

The manuscript needs several important changes and better handling in the coding and interpretation of results, especially at the time of discussion. More information is required in the introductory section that can be perfectly transferred from the paragraphs of the discussion section that I have cited above, and others that can be identified by the authors. It is a strategy to optimize efforts and contents.

In addition, with the suggestions for major changes that I have been indicating for each section, I see a possibility of making the text look at least good enough to justify and defend its publication. In its current state, I do not see it for publication. It needs major changes so that I can reconsider its acceptance.

I hope that my contributions will serve to improve this article and the study you propose. I congratulate you for the effort made. Thank you very much.

Author Response

Reply to the reviewer 2 – ijerph-2168122

We thank the reviewer for all comments that allow us to improve the quality of our work. All questions have been carefully analyzed, and we have tried to fully respond to what was requested. All changes were highlighted in grey.

TITLE AND ABSTRACT

a) The focus group is conducted through the online system, therefore, it is important to indicate this condition in the title and abstract à an online focus group. Indicate in the abstract that the patient is simulated.

Response: Thank you for pointing this out. We agreed with all the comments, and we changed the text accordingly in the revised version of the paper (lines 2-3 and 30-33).

 1 - INTRODUCTION

b) The introduction needs more development, you can move some paragraphs from the discussion section to this section (see my suggestions in discussion).

c) I am struck by the fact that they only use the definition of stigma proposed by Goffman (1963). I suggest reading:

Rossler, W. The stigma of mental disorders: A millennia-long history of social exclusion and prejudice. EMBO Reports, 2016.17(9); 1250-1253. https://doi.org/10.15252/embr.201643041

It may help in a synthetic and summarized way to provide ahistorical overview of the use of the STIGMA concept and the studies that have been conducted. I mean by this that it is important to justify why Goffman's theory is used and not others. Therefore, it is convenient to address the evolution of the concept and reflect it in the article.

Response: Thank you for your suggestions. We developed the introduction using some of the paragraphs from the discussion, highlighted in grey (lines 51-90 and 130-137).

The recommended article by Rossler was integrated into the introduction. Also, the historical overview of stigma was integrated, based on Goffman´s theory of stigma attributes and updated using actual references. Please see the following paragraphs in the introduction section.

“Discrimination has been seen as a behavioral response caused by negative attitudes [3]. Various forms of discrimination can occur and materialize in differential treatment for certain individuals because they belong to specific groups [4].

Prejudice generally refers to attitudes, emotions, or behavior towards group members, which implies, directly or indirectly, some negativity or antipathy towards that group [5]. Prejudice adds emotional content to affective content (stereotypes), giving rise to an unfavorable attitude [6].

Stereotypes are preconceived opinions and attitudes about members of certain groups (e.g., ethnic, religious, mentally ill). The general population has negative and stigmatizing attitudes toward mental health patients, and a lot of myths and stereotypes remain [7-9]. Even though the stereotypes change according to the disease, the dangerousness, unpredictability, and guilt are common and frequently result in negative attitudes and discrimination toward people with mental health problems [10, 11].

Stigma is distinguished from discrimination, prejudice and stereotypes, and it is described in three conceptual domains (i.e., cognitive, emotional, and behavioral) [12], which can have detrimental effects on the health and well-being of people with mental illness [13]. There are several manifestations resulting from stigma, whether individual or collective. Stigma as a threat to identity occurs when an individual evaluates the demands imposed by a stressor of a stigmatizing nature as potentially threatening to their social identity and well-being [14]. Labeling and identity theories explain the development and maintenance of stigmatized identities. The person with mental illness loses their previous or desired identity(s) and starts to adopt stigmatizing views of themselves, limited to their condition, as useless and incompetent. Thus, the impact of the diagnosis on these people's identity of 'being mentally ill' can become dominant in the lives of some individuals [14].

Stigma is defined by Goffman as an “attribute that is deeply discrediting, reducing someone “from a whole and unusual person to a tainted discounted one” [15, p.3]. Subsequent work agrees with this definition [16] and reinforces that stigma is firmly rooted in cultural and contextual issues that differ over time and across contexts [17]. Since then, the definition of stigma has evolved as a product of socialization. Thus, two types of stigma have been underlined: the public stigma and the self-stigma [18].

The term "public stigma" refers to the stigmatizing perceptions that the general public holds of those with mental illnesses. The internalization of public stigma constitutes self-stigma [18], when people with mental illness internalize public attitudes and then suffer low self-esteem, and low self-efficacy, contributing to the compromise of recovery potential [12]. Internalized stigma can impede the effectiveness of treatment, and the recovery process, as people with mental illness experience feelings of shame and self-devaluation and subsequently withdraw from social activities [19]. Family members of the person with mental illness are also affected by internalized stigma; they feel ashamed and blamed for their relative's condition which can lead to isolation and economic difficulties [12].

(…)

“While different measurement tools quantify mental health stigma, we believe there is still a lack of a holistic, comprehensive approach to this issue in nursing students, which is crucial to designing the strategies to curtail barriers to mental health practice, promote help-seeking behaviors, and expand mental health nursing education. To address this gap, this study's primary goal is to understand senior undergraduate nursing students' perspectives regarding mental health stigma and discrimination. For this purpose, it was intended to understand the view of a group of nursing students about mental health stigma starting from a simulated case vignette of a person with a mental health problem.”

2.- MATERIAL AND METHODS.

2.1.- Study design:

d) They should indicate the motive or reason for not conducting the Focus Group in person, Sample selection and recruitment: Indicate the number of students who were in the 4th year, and also specifically the nursing school (Ribeiro Sanchez or others??).

Response: Thank you for sharing your perspective. We add rational for using the online focus group and more detailed information about sample selection and recruitment process. After careful analysis, the research team decided to avoid naming the Nursing School to guarantee the anonymity and confidentiality of the institution and data collected.

Please see the following text (Lines 145-162):

“Three FG were run, with 19 nursing students (9 in the first group, 6 in the second, and 4 in the third FG) from 38 potential students in the 4th year of the bachelor's degree at one Portuguese nursing school. It is generally accepted that between four and twelve participants are sufficient for each FG [35]. The participants were divided according to availability, considering that some were working students.” (…)

“Participants were selected using a purposeful sampling procedure based on previous clinical experience with a person with a mental illness. The inclusion criteria are as follows: (1) adult nursing students (≥18 years old); (2) can communicate in Portuguese; and (3) agree to participate in this study.” (…)

The option of conducting online FG was due because during the period in which they were held, there were no classes, and, in this way, we tried to ensure low dropout rates.  In addition, the threat imposed by the COVID-19 pandemic further strengthened the rationale for conducting FG research online [37].”

e) Data Collection: Indicate if the participants answered with the video image activated, and if there was any protocol to control the attention and participation of the students.

Response: Once again, thank you for pointing this out. During the data collection, all participants had the video cameras and audio activated to ensure the visualization of the nonverbal cues in their interactions. This information was negotiated before each focus group started.  (Lines 168-171)

f) They must clearly indicate that "John" is a simulated patient (line125).

Response: Thank you for your comment. In fact, the vignette presented is a simulated case-based scenario. This information was integrated into the abstract and in the main text.

g) On line 132 they should indicate the type of interview: structured, semi-structured, etc....

Response: This inconsistency was replaced in the text. We used a semi-structured interview guide to conduct all the focus groups (lines 181-184 and table 1).

“The data was collected using a semi-structured interview guide with open questions, which allowed us to get to know the participants' points of view better, and to get a more explicit idea of their perceptions and experiences.”

2.5.- Data Analysis

  1. h) They should indicate the advantages or reasons for choosing Braun and Clarke's model and not that of other authors, e.g. Smith IPA.

https://www.google.com/url?sa=t&rct=j&q=&esrc=s&source =web&cd=&ved= 2ahUKEwjXz7GisJL9AhUgXaQEHcEGCNQQFnoECA0QAQ &url=https%3A%2F%2Fmed-fom-familymed-research.sites.olt.ubc.ca%2Ffiles%2F2012%2F03%2FIPA_Smith_

Osborne21632.pdf&usg=AOvVaw1svhzlgxwzZXODOdivaV_x

I respect your choice, but considering the objective and content of the manuscript, I think Smith's model is more appropriate.

i) Please indicate why you do not use phenomenological analysis(IPA) in the Focus Group.

Response: Thank you for your insightful contribution. Despite the Smith model could be a good and interesting choice, and after a careful discussion, we decided to be consistent with the thematic analysis proposed by Braun and Clarke. This approach is justified because it allows a deductive “top-down” way or inductive 'bottom-up' way to explain themes, highlight similarities and differences, and generate unanticipated insights. (Lines 196-199)

j) In lines 156-157 I miss the Kappa indicators that numerically reflect the level of consensus on the three main themes identified.

Response: Thanks for noticing this issue. Interrater reliability was calculated using the percentage of agreement the Polit and Beck, (2012). Thus, we considered a percentage value equal to or greater than 90% to consider that there was a consensus. (Lines 214-216)

“Interrater reliability was calculated using the percentage of agreement [40]. Thus, we considered a percentage value equal to or greater than 90% to consider that there was a consensus.”

3.RESULTS

 l) Sample description

In lines 190 to 192 it is stated that there are no previous diagnosis of mental illness and in table 2 it is stated that there are four cases. The level of mental health is defined as good or very good, and in the table there are 7 cases of very bad to regular levels.

Response: Thank you for your perspicacity. We corrected the participant’s information regarding the presence of mental symptoms.  In fact, they do not have a diagnosis of mental illness, but at the moment of data collection, they revealed the presence of mental symptoms (anxiety and depression. n=4).  (Line 250-252 and table 2)

m) The composition of FG1, FG2 and FG3 is not described in detail. Given how small the sample is, and that it is not reported why the groups are not balanced with the same or similar number of participants, they should make a detailed table with the same variables used in Table 2, but distributing by FG1 FG2 and FG3. This may be interesting to understand the level of depth with which the issues are addressed in the different FGs.

Response: Thanks again for your comments. We detailed the information regarding each FG and presented them in table 3 revised. (Please see the table 3 in the revised document). The distribution of the participants by each focus group was determined by the availability of the students to participate in each focus group. 

 3.2. Themes and sub-themes emerged.

n) Four sub-themes and 14 different codes are mentioned. If I am not mistaken, Table 3 cites 6 sub-themes, and 16 codes.

Response: Thank you again for your comment. In order to improve the readability of the table we include the bullet points and lines to the table.  Thus, we have 3 themes, 5 sub-themes and 14 codes. (Table 3)

o) In the coding of the literal discourse (3.2.1, 3.2.2. and 3.2.3) the same format in italics should be used. Please note that at other times you use only quotation marks without italics. At the end of each discursive quotation, you should indicate the group and the code of the interviewee and sex, for example FG1-XX-1, or FG3-XY-4, etc....

Response: Thank you and we agree with your suggestions. We add the same format to the code extract and add the alpha numeric numbering in each extract sample.

Example extracts from the participants were numbered according to the FG number (1–3), by sex (XX/XY), and by interviewee number.

p) In section 3.2.1., stereotypes, FG2 is underrepresented, there is no expression of these participants. The same in Prejudice, FG1is omitted, and in section 3.2.4, the opinions of group FG3 are not taken into consideration. The three FGs must be given a voice in some way, this underrepresentation cannot occur, with the problem that the three FGs are unbalanced with different numbers of participants.

Response: Thank you for your comments. We add more quotes to highlight the themes and subthemes we found. These are extended to all themes.

4. DISCUSSION

q) This is a very extensive discussion, which contains elements that could well be moved to the previous sections. Lines 349 and 350 can fit in the procedure section. Lines 378 to 381 (prejudice) can fit in the introduction. Limit the discussion only to commenting on the results with reference to existing results in other studies.

Response: We agree with our comments, and we reorganize the discussion section.  We also try to integrate the previous literature to explain the response patterns of our findings. 

r) You say that interaction among participants is a strength of the FG, but this is more so when it is approached face-to-face. Here the format is online, and this entails some very important limitations that the authors do not indicate. It should be clearly stated what the problem with online FGs is.

Response: OK! We introduce some cons and pros of the online focus group in our limitation subsection. 

We hope we have accomplished all the suggestions to consider this paper to be published. Thank you again for helping us to improve our work.

Yours sincerely

The authors

Round 2

Reviewer 2 Report

I would like to thank the editors and authors for the opportunity to review again the article LOOKING BEYOND MENTAL HEALTH STIGMA: A FOCUS GROUP STUDY AMONG NURSING STUDENTS.

 In general, I can confirm that the authors have re-drafted the manuscript in response to my modest suggestions for improvement.

The current version of the article presents a more reasoned introduction that justifies the study from the chosen theoretical model. The incorporation of some paragraphs of the discussion to the introduction section, with the requested clarifications, makes the background more consistent with the objectives pursued.

In the method section, authors introduce complementary information to facilitate the replication of the study. In addition, the results are presented in a clearer and more detailed manner, including a better codification of the FG informants.

The introduction and discussion in view of the results obtained, is better connected due to the introduction of bibliographic references oriented to the differentiation of the different variables used that come from the social psychology of prejudice and stigma.

Finally, the limitations section is much more realistic and consistent with the type of design used.

I note that the authors have made an important effort and work following my suggestions for improvement.  I honestly believe that the manuscript in its current situation meets the minimum conditions for publication.

Therefore, my current proposal is to ACCEPT it for publication.

I would like to congratulate the authors for the work done and for their professional behaviour. Also for their respectful tone and kindness in understanding my proposals for improvement.

Best regards